# Baicalin Relieves LPS-Induced Lung Inflammation via the NF-κB and MAPK Pathways

**DOI:** 10.3390/molecules28041873

**Published:** 2023-02-16

**Authors:** Bingyu Shen, Haoqing Zhang, Zhengjin Zhu, Zixi Ling, Fangyuan Zeng, Yazhou Wang, Jianguo Wang

**Affiliations:** College of Veterinary Medicine, Northwest A&F University, Yangling 712100, China

**Keywords:** baicalin, ALI, cytokines, NF-κB

## Abstract

Baicalin is an active ingredient extracted from the Chinese medicine *Scutellaria* and has many beneficial effects. Pulmonary interstitial and alveolar edema are common symptoms of an acute lung injury (ALI). We investigated the effects of baicalin on LPS-induced inflammation and the underlying mechanisms in mice and cells. The protein contents and mRNA expression of TNF-α, IL-1β, and IL-6 in RAW264.7 cells and mice were detected using ELISA and qRT-PCR. Baicalin significantly suppressed TNF-α, IL-1β, and IL-6 levels and expression, both in vitro and in vivo, compared with the LPS group. Baicalin inhibits the expression of TLR4 and MyD88, resulting in significant decreases in p-p65, p-p38, p-ERK, and p-JNK, as measured by the Western blotting of RAW264.7 cells. A baicalin treatment for 12 h resulted in a rapid increasing of the white blood cell number and significantly improved the pathological changes in the lung. We also found that the baicalin pretreatment for 12 h could decrease the MPO content and wet/dry (W/D) weight ratio, which indicates that baicalin can significantly reduce pulmonary edema. Furthermore, the baicalin pretreatment also resulted in the recovery of TGF-β protein levels and decreased iNOS. Baicalin inhibits ALI inflammation in mice and cells and is a potential candidate for the treatment of ALI.

## 1. Introduction

Acute lung injury (ALI) is an excessive or uncontrolled lung disease caused by direct or indirect lung injury. It is characterized by pulmonary edema and acute inflammation that often develops into a more severe acute respiratory distress syndrome. Currently, LPS is commonly used to establish models of ALI, where it is injected through nasal and trachea drops or intraperitoneally, resulting in the increased accumulation of peripheral inflammatory cells, enhanced pulmonary capillary permeability, increased alveolar and interstitial edema, and ultimately an acute inflammatory response [1,2,3], leading to tissue damage or shock. When the lung tissue is injured, neutrophils, in addition to macrophages, rapidly accumulate in pulmonary microvessels after activation, leading to the release of toxic substances, including proteases, reactive oxygen species, and proinflammatory cytokines, which eventually leads to increased vascular permeability [4]. Gram-negative bacteria produce lipopolysaccharides (LPS) as part of their outer cell wall, and this initiates an inflammatory response [5]. LPS can interact with the LPS binding protein (LBP) to significantly enhance the LPS activity and induce a small amount of LPS to become bioactive [6]. In addition, the LPS–LBP complex is transferred to CD14 on the cell surface [7], where it can activate numerous receptors such as toll-like receptors (TLR), mitogen-activated protein kinases (MAPK), and the nuclear factor κB (NF-κB) [8]. The transcription of a number of important pro-inflammatory cytokines, including the tumor necrosis factor (TNF)-α, interleukin (IL)-6, IL-1β, and NO, is driven by the resulting cascades [9]. The LPS-induced cytokine expression is a key event in activating the immune system. For instance, when LPS is transferred into blood, it stimulates cells via TLR-4 and NF-κB to activate a series of immune responses [10]. Therefore, LPS can be intranasally administered to mice to establish an ALI model, as LPS damages lung cells by recruiting inflammatory cells.

NF-κB is a universally distributed transcription factor, and it participates in regulating the expression of genes involved in the cell cycle, immune response, and DNA repair. Antigens and mitogens, as well as cytokines and mitogens, can trigger this process [11]. A major role has been established for the NF-κB in ALI pathogenesis [12]. MAPK family members, including JNK, ERK1/2, and P38 MAPK, play essential roles in oxidative stress-induced apoptosis, other forms of apoptosis, and mitochondrial dysfunction [13,14,15,16]. Thus, it is essential to investigate the mechanisms of baicalin on NF-κB and MAPK signaling pathways.

Baicalin is a major active ingredient isolated from *Scutellaria baicalensis,* a type of Chinese herb that is utilized widely in Chinese traditional medicine. Baicalin has been demonstrated to have various biological activities, for instance, cardioprotective, anticancer, antiproliferative, antioxidant, and anti-inflammatory effects [17]. The research of Huang et al. indicated that baicalin exerts protective effects on the rabbit in resisting articular chondrocytes [18]. According to Chen et al., baicalin inhibits HMGB1/RAGE and modulates pulmonary hypertension in an infant rat model [17]. A few studies have reported the effects of baicalin on lung injury, but the specific mechanism is currently unclear.

In our study, we investigate whether baicalin can decrease the lung inflammation in mice and in RAW264.7 cells by detecting changes in the expression of NF-κB and MAPK proteins. This would not only provide basic evidence in the context of acute lung injury but also indicate whether baicalin can potentially be used in clinical application in future treatments.

## 2. Results

### 2.1. In Vitro Study

#### 2.1.1. Baicalin Had No Effect on Cell Viability

The RAW264.7 cells treated with baicalin (60, 90, 120, 150, and 180 μg/mL) showed no differences in cell viability compared with the control groups, regardless of whether the LPS was absent or present (Figure 1B,C). Nontoxic and efficient concentrations of baicalin (60, 90, and 120 μg/mL) were used in the following in vitro experiments.

#### 2.1.2. Baicalin Inhibits Inflammatory Cytokine Production in RAW264.7 Cells

The effects of baicalin on the generation of inflammatory cytokines were determined using qRT-PCR and ELISA. Compared with the control group, LPS observably enhanced the contents of TNF-α, IL-1β, and IL-6, whereas their levels were remarkably lower in the baicalin-treated groups (60, 90, and 120 μg/mL). The results of qRT-PCR show the same dose-independent tendency as those of ELISA, which clearly demonstrate the inflammation-relieving effects of baicalin (Figure 1D–I).

#### 2.1.3. Baicalin Inhibits NF-κB and MAPK Activation of RAW264.7 Cells

Baicalin’s anti-inflammatory properties were investigated by Western blotting to investigate how it affects NF-κB and MAPK signaling pathways. From Figure 2 and Figure 3, we find that after LPS induction, the levels of IκBα, p65, ERK, JNK, and p38 phosphorylation were obviously increased. Pretreatment with baicalin after LPS induction suppressed this upregulation of IκBα, p65, ERK, JNK, and p38 phosphorylation compared with the LPS group. Meanwhile, the contents of TLR4 and MyD88 were observably decreased after the addition of baicalin (Figure 3G). Furthermore, from immunofluorescence, we can draw the conclusion that baicalin inhibits the transportation of NF-κB p65 to the nucleus (Figure 2F).

### 2.2. In Vivo Study

#### 2.2.1. Baicalin Decreases Cytokine Levels in LPS-Induced ALI Mice

As demonstrated in Figure 4A–C, the production of cytokines (TNF-α, IL-1β, and IL-6) were significantly augmented after the LPS challenge compared with the control groups. The baicalin pretreatment demonstrated a more marked and dose-independent decrease in the contents of pro-inflammation cytokines than the LPS group.

#### 2.2.2. Baicalin Reduced the Increase Inflammatory Cells Induced by LPS

We determined changes in the number of various cells in the blood of mice following the baicalin treatment for 6, 12, and 24 h. The results demonstrated that compared with the control group, the number of WBCs (including lymphocytes, monocytes, and neutrophils) increased significantly at 12 h after the LPS induction, while the baicalin treatment decreased the enrichment of WBCs and other inflammatory cells compared with the LPS group. Meanwhile, the baicalin treatment for 12 h resulted in the strongest reduction in inflammatory cells compared with the treatment at 6 and 24 h (Table 1 and Appendix A).

#### 2.2.3. Baicalin Reduced Lung Tissue W/D Ratio

The wet to dry ratio (W/D) intuitively indicates the degree of edema in lungs. As observed in Figure 4D, administering the LPS to mice resulted in a dramatic rise in the W/D ratio, which represents the level of the amelioration of lung edema. However, the baicalin inhibited this rising tendency by reducing the ratio.

#### 2.2.4. Baicalin Alleviates Lung Tissue Histopathological Changes in Mice Caused by LPS

The histopathological alterations in lung tissue were examined by microscopy after H&E staining. The lung of the LPS induced for 6 h, 12 h, and 24 h showed that the alveolar septum was thickened, the alveolar wall telangiectasia was congested, and a large number of inflammatory cells were infiltrated near the bronchi (deep blue purple). Neutrophils and macrophages could be observed in the alveolar space. After the baicalin treatment for 6 h, 12 h, and 24 h, the infiltration of inflammatory cells was significantly reduced, and the alveolar structure was intact. The degree of the inflammatory cell infiltration, capillary congestion, hemorrhage and alveolar wall thickening were gradually and obviously reduced in the baicalin treatment for the 12 h groups (Figure 5).

#### 2.2.5. Baicalin Reduces α-1 AT, IgE, and MPO and Increases IgG in LPS-Induced ALI Mice

The results demonstrated that after the LPS was induced, the content of the α-1 AT increased significantly compared with the control group, and the application of baicalin (25, 50, and 100 mg/kg) reduced the increasing content of α-1 AT and demonstrated a dose-dependent manner (Figure 6A). Meanwhile, the LPS-treatment induced mice produces a higher IgE level compared with the control group, while the baicalin significantly reduced the production of this allergen in a dose-dependent manner (Figure 6B). Mice in the control group had a higher level of IgG compared with the LPS group. Three concentrations of baicalin (25, 50, and 100 mg/kg) restored the IgG level gradually (Figure 6C). The biological action of MPO serves as a functional marker and sign of neutrophil activation. In Figure 6D, the increasing MPO levels in the LPS group indicate the activation of neutrophils and macrophages, whereas the pretreatment with baicalin inhibits the MPO activity, which indicates that baicalin reduces the accumulation of total cells and macrophages (Figure 6D).

#### 2.2.6. Baicalin Decreased the Protein Expression of IκB, iNOS, and TGF-β in LPS-Induced ALI Mice

An immunohistochemical technique was applied to detect the expression of IκB, iNOS, and TGF-β. The results demonstrated that after the LPS was induced, the protein expression of IκB, iNOS, and TGF-β were increased compared with the control mice. The baicalin pre-treatment reduced the ascending protein expression (Figure 7).

## 3. Experimental Section

### 3.1. Cell Culture and Animal Feeding

The RAW 264.7 macrophage cell line was donated by China Cell Line Bank (Beijing, China), and the culture medium included 10% fetal bovine serum (FBS) and 100 units per milliliter of penicillin and streptomycin, with culturing under 5% CO_2_ and at 37 °C.

Liaoning Changsheng Biotechnology supplied male BALB/c mice weighing between 18 and 22 g (Liaoning, China). Standard food and water were given to the mice kept under SPF conditions (24 ± 1 °C, 40–80% relative humidity). All experiments followed the National Institutes of Health (NIH) guide for the Care and Use of Laboratory Animals and were approved by the NWAFU Animal Administration Committee (No.2021054).

### 3.2. Chemicals and Reagents

Baicalin (HPLC ≥ 98%) was purchased from Shanghai Yuanye Bio-Technology. Fetal bovine serum (FBS), Dulbecco’s modified Eagle’s medium (DMEM), penicillin, and streptomycin were provided by Gibco. LPS (*Escherichia coli* 055:B5) and the cell counting kit (CCK-8) were purchased from Sigma. Mouse TNF-α, IL-6, and IL-1β ELISA kits were purchased from Biolegend. The antibodies against phospho-IκBα, IκBα, NF-κB p65, and phospho-p65 were purchased from Abcam. Phospho-p38, p38, p-JNK, JNK, p-ERK, ERK, TLR4, and MYD88 were purchased from Cell Signaling Technology. Pyrrolidine dithiocarbamate (PDTC, an NF-κB inhibitor), SP600125 (a JNK inhibitor), SB203580 (a p38 MAPK inhibitor), and U0126 (an ERK inhibitor) were provided by Selleck. HRP-conjugated goat anti-rabbit and goat-mouse antibodies were supplied by BOSTER.

### 3.3. In Vitro Study

#### 3.3.1. Cell Ability Assay

RAW 264.7 cells (4 × 10^5^ cells/mL) were cultivated in 96-well plates for 12 h, followed by a 1 h treatment with different concentrations of baicalin (60, 90, 120, 150, and 180 μg/mL) and 18 h of LPS stimulation (4 μg/mL, 50 μL/well). Each experimental group received a 10 μL CCK-8 kit with incubation for 3 h. A microplate reader was used to measure the optical density at 450 nm (TECAN, Australia).

#### 3.3.2. Cytokines’ Assay

RAW 264.7 cells (4 × 10^5^ cells/mL) were cultivated in 24-well plates for 12 h and then treated with baicalin (60, 90, and 120 μg/mL) or PDTC (10 μM). After 1 h, LPS (4 μg/mL) was added to the LPS and LPS + baicalin groups with incubation for another 18 h. The TNF-α, IL-1β, and IL-6 levels were measured in the cell culture supernatant using ELISA kits (Bio Legend).

#### 3.3.3. Western Blot Analysis

RAW 264.7 cells were cultivated in 6-well plates to an initial density of 8×10^5^ cells/well for 24 h and treated with baicalin (60, 90, and 120 μg/mL). The other three groups were treated with PDTC (treatment concentration of each substance is 10 μM) for 2h. Aside from the control, LPS (4 μg/mL) was added to all other groups with incubation for 18 h. Scraped cells were centrifuged in EP tubes at 3000 rpm for 10 min at 4 °C. A RIPA lysis buffer containing protease inhibitors and phenylmethanesulfonylfluoride (PMSF) was added to sediments for 30 min on ice. The mixtures were centrifuged at 12,000 rpm for 10 min at 4 °C. The supernatants were collected, and the concentration of protein was detected using a BCA protein assay kit (Beyotime, China). Depending on the protein concentration, 5× loading buffer and the appropriate amount of PBS solution were added to adjust the quantity of each sample to 80 μg in a volume of 20 μL for adding to wells of 10% SDS-PAGE gels. Proteins were transferred to PVDF membranes, which were then placed into 5% skimmed milk solution with shaking for 1 h. The primary antibodies were diluted using TBST (1:1000) at 4 °C overnight. HRP-conjugated antibodies were diluted in TBST (1:5000) with incubation for 45 min at room temperature. The ECL Western blotting detection system was used to detect immunoreactive proteins (Pierce, IL, USA).

#### 3.3.4. Quantitative Real-Time Polymerase Chain Reaction

The Primer Express 5.0 software was used to create the exact primers for NF-κB p65, IκB, IL-6, TNF-α, IL-1β, and β-actin based on the sequences that are known (Table 1). TRIzol was used to extract the total RNA from RAW 264.7 cells. A K5500 Micro-Spectrophotometer was used to measure RNA concentrations and purity. PrimeScript Reverse Transcriptase was used to reverse-transcribe 5 ng of total RNA into cDNA (TaKaRa). Using a SYBR Green QuantiTect RT-PCR Kit (Takara Biotechnology Co., Ltd., Kusatsu, Japan), a quantitative polymerase chain reaction (qRT-PCR) analysis was used to examine the mRNA expression level of NF-κB p65, IκB, IL-6, TNF-α, IL-1β, and β-actin. A 7500 Real-Time PCR System was used to implement qRT-PCR (Applied Biosystems). The relative expression of each gene was normalized based on the levels of β-actin.

#### 3.3.5. Immunofluorescence

Glass slides were treated as follows: soaking overnight in acid alcohol (75% alcohol containing 1% HCl acid), immersion in distilled water for 2 h, bubbling for 36 h, flushing with water 3 times, ethanol soak overnight, and dry roasting (180 °C, 3 h). RAW 264.7 cells (4 × 10^5^ cells/mL) were cultivated in 24-well plates with glass and treated with baicalin (60, 90, and 120 μg/mL). After 24 h, they were fixed in 4% formalin for 20 min at room temperature, and the slides were then incubated with EDTA_2_Na for 2 h (boiling water, 5 min). Afterwards, 0.1% Triton was used to permeabilize the samples on slides for 10 min, and then the samples were incubated overnight at 4 °C with the NF-κB p65 antibody. After further washing, the slides were exposed to the goat anti-rabbit IgG labeled with cy3 for 30 min. After being sealed with glycerin in a dark place at 4°C, the slides were examined using a laser scanning confocal microscope after being stained with DAPI.

### 3.4. In Vivo Study

#### 3.4.1. Mice Model

The male mice were randomly divided into 5 groups (*n* = 6): control group; LPS (5 mg/kg) group; LPS + baicalin (25, 50, and 100 mg/kg). The baicalin was intraperitoneally injected. After 1h, the mice of the LPS group and the LPS + baicalin group received 50 μL LPS intranasally, while the control group received PBS intranasally.

#### 3.4.2. Blood Routine Examination

The blood sample of all mice was collected for determining the CBC, differential leukocyte counts, and other mediators after the baicalin treatment for 6 h, 12 h, or 24 h (6 h and 24 h treatment concentration were 100 mg/kg), respectively. Blood samples were mixed with EDTA_2_K and measured on a Mindray Animal Automatic Blood Cell Analyzer (BC-2800vet).

#### 3.4.3. BALF Collection and Cytokine Assay

Bronchoalveolar lavage fluid (BALF) was collected from all groups after 12 h using a tracheal cannula containing PBS. ELISA kits were utilized to measure the concentrations of TNF-α, IL-6, and IL-1β according to the manufacturer’s protocol.

#### 3.4.4. Lung W/D Ratio

The weights of fresh wet lungs of baicalin treatment for 12 h were measured immediately after euthanizing the mice. The lungs were then placed in an 80 °C oven to dry for 48 h and weighed again. The W/D ratio was calculated by dividing the wet weight by the dry weight.

#### 3.4.5. Histopathologic Evaluation of the Lung Tissue

Mice were treated with LPS or/and baicalin for 6 h, 12 h, and 24 h. Lungs were collected after the mice were executed and then fixed in a 10% formaldehyde solution. Light microscopy revealed pathological changes in the lung tissues of mice after dehydrating until transparency and paraffin embedding and dewaxing.

#### 3.4.6. Alpha-1 Antitrypsin (α-1 AT), IgE, IgG, and Myeloperoxidase (MPO) Activity Assay

Accurately weighed lung tissue samples were ground into homogenate. The α-1 AT, IgE, IgG and MPO activity was measured using ELISA kits according to the manufacturer’s protocols.

#### 3.4.7. Immunohistochemical of Lung Tissue

Lungs from groups of mice treated for 12 h were collected for determining the expression of IκB, TGF-β, and iNOS. The lungs were first fixed in paraformaldehyde, then embedded, sectioned, deparaffinized, antigen repaired, blocked, and incubated with the primary antibody overnight at 4 ℃. On the second day, the sections were incubated with secondary antibodies and with added chromogen, then stained with hematoxylin, dehydrated, sealed, and observed under a microscope and photographed.

#### 3.4.8. Statistical Analysis

A standard error of mean (SEM) of each value was calculated. One-way ANOVA (Dunnett’s *t*-test) and Student’s *t*-tests were used to assess the differences between the mean values of the normally distributed data. The significance was defined as *p* < 0.05 or *p* < 0.01.

## 4. Discussion

The characteristics of ALI include damage to the intactness of the endothelial and epithelial cells of pulmonary edema, which widely promotes inflammatory mediators and neutrophil infiltration, as well as shock, sepsis, and ischemia [19]. Patients with a critical illness are at risk for death and morbidity due to this inflammatory lung disease. Importantly, the destruction of the alveolar–capillary barrier and an alveolar epithelial injury play a vital role in this type of injury [19]. The most common risk factor for ALI is severe sepsis due to bacterial or viral pneumonia.

LPS is an important factor causing pneumonia, and it participates in the pathogenesis of pulmonary edema. TLRs are a central element of immune responses through recognizing the different molecules associated with pathogens. Their primary mechanism of activation involves MyD88 and/or other proteins, resulting in inflammatory cytokine production [19]. The control of the MAPK and NF-κB inflammatory reaction is advantageous for maintaining an endothelial barrier integrity. The LPS stimulation activates neutrophils that migrate from the blood circulation to the lung interstitium, resulting in a significant increase in ROS production, which further accelerates the inflammation [20]. These results are achieved through the inhibition of MAPK and NF-κB in addition to other inflammatory effects [20].

Lipopolysaccharide, as an endogenous and exogenous factor, has been extensively studied due to its potential to cause endothelial barrier dysfunction [21]. In Gram-negative bacteria, it binds to the TLR4, which activates signaling pathways [22,23]. Oxidative stress, inflammation, and endothelial damage are further induced by LPS-activating macrophages, neutrophils, and other cell types [24]. As a result of exposure to LPS, the endothelial barrier becomes dysfunctional and permeability increases [25]. Excessive oxidative stress and/or inflammatory responses could also aggravate the pathogenesis of ALI [26,27]. In the current study, we investigated whether baicalin could have an effect on the LPS-induced ALI-mediated NF-κB signaling pathway conferring anti-inflammatory properties. As the first line of defense against bacteria and fungi, neutrophils play an important role in the innate immune response. However, neutrophils also play an important role in the tissue damage of acute disease processes, such as acute lung injury [28]. LPS induced the highest levels of white blood cells, including lymphocytes, neutrophils, and monocytes, in the blood of mice at 12 h, compared with at 6 and 24 h. In addition, the baicalin treatment for 12 h significantly reduced the enrichment of WBCs, mainly neutrophils, in blood.

Immune responses, the cell cycle, and various cells and tissues are regulated by NF-κB, a universal transcription factor [29]. The activation and inhibition of NF-κB is the core contributing factor of many diseases; thus, NF-κB has become the main focus of therapeutic interventions [11]. The NF-κB activation mediates an increased regulation of several pro-inflammatory cytokines, resulting in the occurrence of inflammation and a subsequent cascade of reactions [30,31]. In the study of Xudong Sun et al., histamine could significantly increase the concentrations and mRNA expression of TNF-α, IL-6, and IL-1β via the activation of NF-κB [32]. According to the results of our research, the production of pro-inflammatory cytokines, such as TNF-α, IL-1β, and IL-6, was remarkably enhanced compared with the control group. In the groups treated with baicalin, the levels of TNF-α, IL-1β, and IL-6 were markedly declined compared with the LPS-only group (Figure 1 and Figure 4). As shown in Figure 2, Figure 3 and Figure 7, the degree of the IκBα, p65, ERK, JNK, and p38 phosphorylation was significantly enhanced in the LPS group. The pretreatment with baicalin (25, 50, and 100 mg/kg) markedly inhibited this rise in levels. We can draw the conclusion from Figure 3 that baicalin reduces the contents of TLR4 and MyD88. This indicates that the anti-inflammatory impacts of baicalin occur by potentially affecting the activation of NF-κB and MAPK via the TLR4 pathways.

It is well known that in the process of ALI, some pathological changes would accumulate in the lung tissue, such as pulmonary edema and hemorrhage. In the LPS-induced ALI model of this experiment (Figure 5), the lungs experienced extensive damage, with symptoms including interstitial edema, bleeding, and an increase in the alveolar wall thickness, inflammation, and cells infiltrating the reproductive system. Instead, after treatment with baicalin, the pathological lung tissue was alleviated. α1-AT is one of the most important protease inhibitors synthesized by hepatocytes in vivo. Due to its low molecular weight, it is easy to diffuse in lung tissue and combine with destructive proteolytic enzymes, which can prevent the lung tissue from being damaged by proteolytic enzymes, thereby protecting the normal structure and function of the lung tissue. In addition, α1-AT is also an acute phase protein. When the body has inflammation, infection, cachexia, and other conditions, its blood concentration will increase 3 to 4 times. Our results demonstrated that after LPS induced, the content of α-1 AT increased significantly compared with the control group, which illustrated that during the acute lung injury, elastase and trypsin are released in large quantities of mice, resulting in an imbalance of the ratio of protease to α-1 AT. Moreover, the application of baicalin (25, 50, and 100 mg/kg) reduced the increasing content of α-1 AT and demonstrated a dose-dependent manner. IgE is mainly synthesized by B cells in the lymphoid tissue of the lamina propria of the mucosa of the respiratory tract and digestive tract, which is a mediator of allergic reactions. It is a secreted immunoglobulin that is the main antibody causing a type I hypersensitivity and has immune functions capable of binding to mast cells and basophils. Meanwhile, the LPS-treatment-induced mice produce higher IgE levels compared with the control group, while baicalin significantly reduced the production of this allergen in a dose-dependent manner. IgG is the main component of serum immunoglobulin, which plays a protective role in the body’s immunity and responds to most antibacterial and antiviral drugs. It can effectively prevent the corresponding infectious diseases by coping with most bacteria, viruses, measles, hepatitis A, etc. The content of serum IgG was detected simultaneously. The mice in the control group had a higher level of IgG compared with the LPS group. Moreover, three concentrations of baicalin (25, 50, and 100 mg/kg) restored the IgG level gradually. MPO, a heme protein, is abundant in neutrophils and is a functional marker of neutrophil activation that plays a part in many aspects; for instance, it produces and regulates inflammatory responses in the body, which then cause a variety of complications [33,34]. As shown in Figure 5, the pretreatment with baicalin significantly reduced the MPO production and lung neutrophilia caused by LPS.

In conclusion, our results demonstrate that baicalin, both in vivo and in vitro, reduces ALI inflammation by diminishing the accumulation of pro-inflammatory cytokines and inflammatory cells, also downregulating the expression of proteins in the NF-κB and MAPK pathways. These observations strongly indicate that baicalin should be considered a new potential target in the treatment of ALI and other lung diseases.

## Figures and Tables

**Figure 1 molecules-28-01873-f001:**
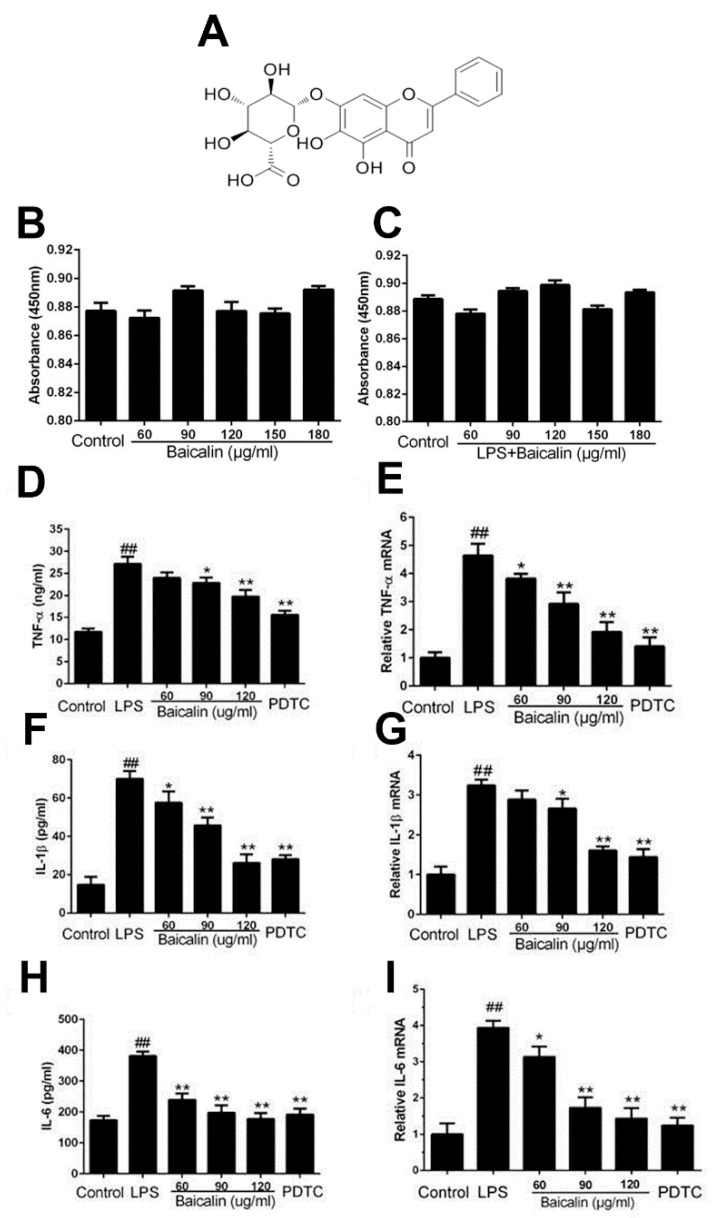
**Effect of baicalin on the cell viability and inflammatory cytokines.** (**A**) Baicalin chemical structure. (**B**,**C**) Effect of baicalin on the cell viability. Cells were cultured with baicalin (0–180 μg/mL) in the absence or presence of LPS (4 μg/mL) for 18 h. (**D**) The content of TNF-α. (**E**) The mRNA level of TNF-α. (**F**) The content of IL-1β. (**G**) The mRNA level of IL-1β. (**H**) The content of IL-6. (**I**) The mRNA level of IL-6. The values represent mean ± SEM of three independent experiments, and differences between mean values were assessed by Student’s *t*-test. ^##^
*p* < 0.01 vs. control group; * *p* < 0.05 vs. LPS: ** *p* < 0.01 vs. LPS.

**Figure 2 molecules-28-01873-f002:**
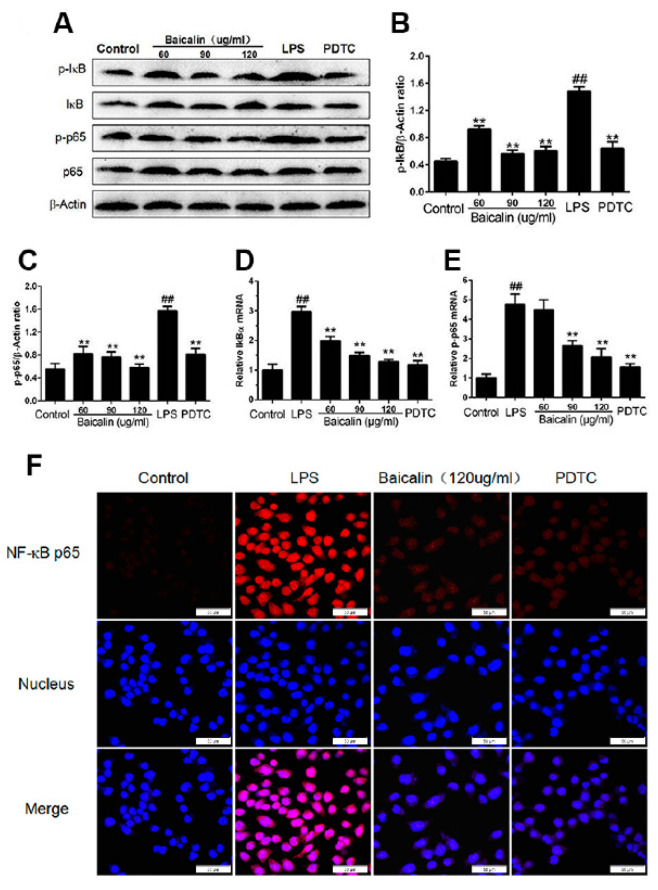
**Effect of baicalin on protein expression NF-κB pathway and NF-κB p65 transcriptional activity of RAW 264.7 cells.** Cells were treated with LPS (4 μg/mL), baicalin (60, 90, and 120 μg/mL), and PDTC (10 μM): (**A**–**C**) Protein expression of p-IκBα, IκBα, β-actin, NF-κB p65, and p-p65; (**D**) Relative mRNA expression levels of p- IκBα normalized to β-actin; (**E**) relative mRNA expression levels of p-p65 normalized to β-actin; (**F**) P65 cellular localization was analyzed by immunofluorescence assay under a laser confocal microscopy, respectively (×400). ^##^
*p* < 0.01 vs. control; ** *p* < 0.01 vs. LPS.

**Figure 3 molecules-28-01873-f003:**
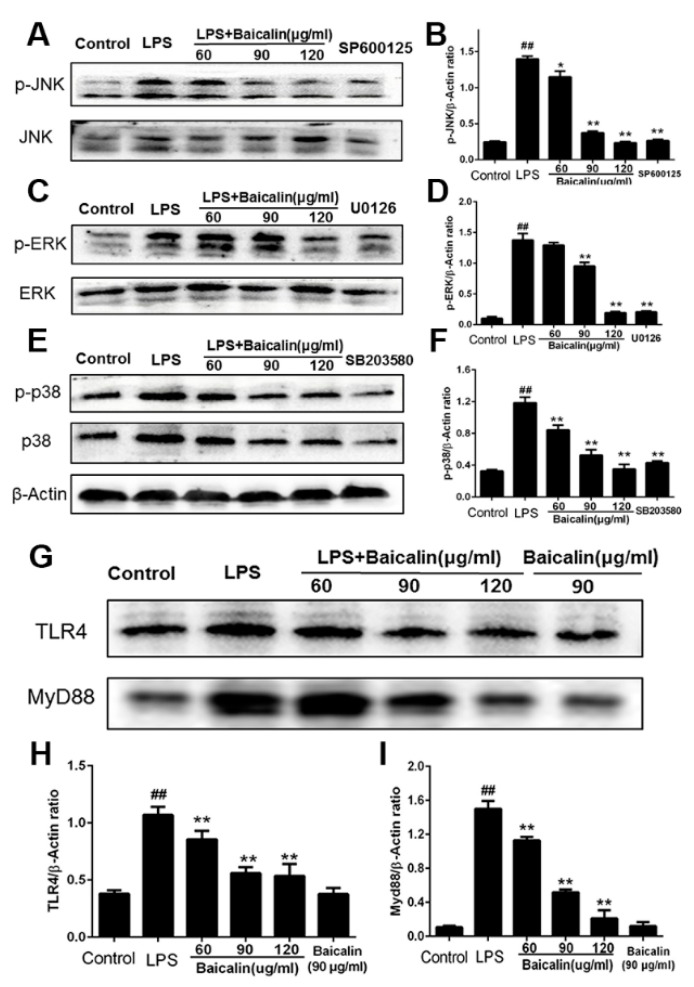
**Effect of baicalin on protein expression in MAPK and TLR4 pathway of RAW 264.7 cells.** Cells were treated with LPS (4 μg/mL), baicalin (60, 90, and 120 μg/mL), SP600125 (10 μM), U0126 (10 μM), or SB203580 (10 μM). (**A**,**B**) Protein expression of p-JNK and JNK; (**C**,**D**) protein expression of p-ERK and ERK. (**E**,**F**) Protein expression of p-p38 and p38; (**G**–**I**) protein expression of TLR4 and MyD88. ^##^ *p* < 0.01 vs. control; * *p* < 0.05 vs. LPS; ** *p* < 0.01 vs. LPS.

**Figure 4 molecules-28-01873-f004:**
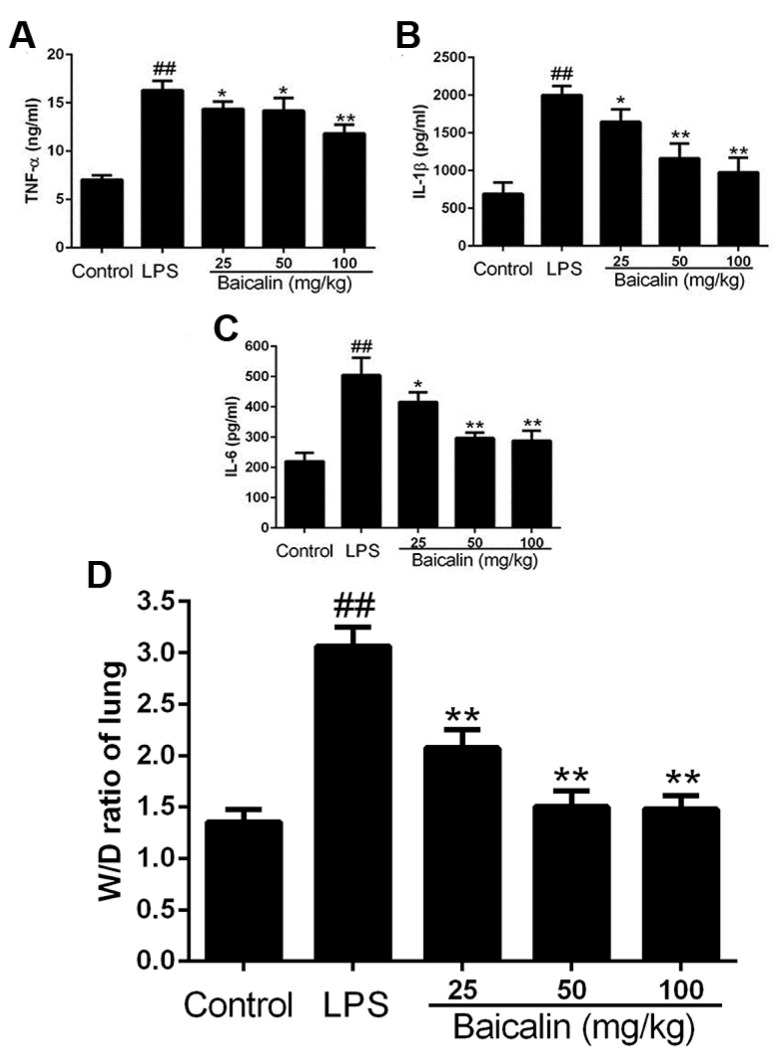
Effect of baicalin on the TNF-α, IL-1β, and IL-6 levels in BALF and lung wet-to-dry weight (W/D) ratio of LPS-induced ALI mice. LPS + baicalin group mice (*n* = 6 per group) were given an intraperitoneal injection of baicalin 1 h prior to administration of LPS. After 12 h, BALF was harvested for the analysis of cytokine levels. The values presented are the means ± SEM. (**A**) The content of TNF-α; (**B**) the content of IL-1β; (**C**) the content of IL-6; (**D**) the W/D ratio of lung in LPS- and baicalin-induced mice. ^##^ *p* < 0.01 vs. control; * *p* < 0.05 vs. LPS; ** *p* < 0.01 vs. LPS.

**Figure 5 molecules-28-01873-f005:**
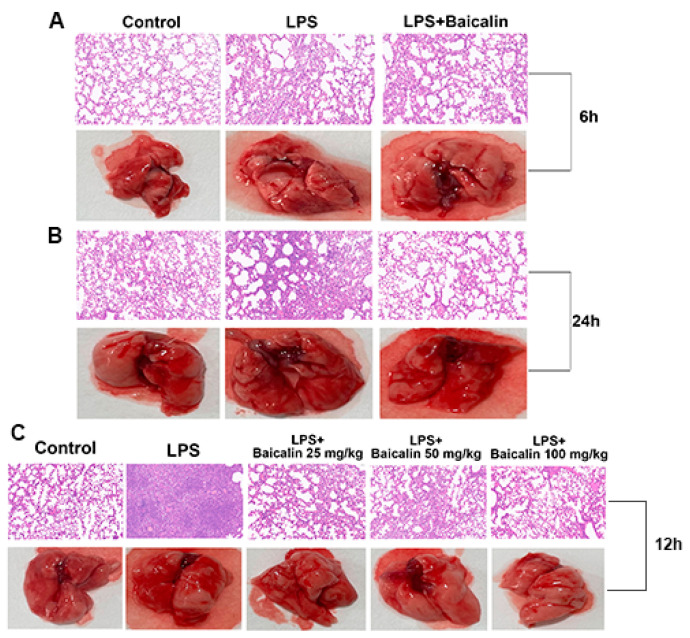
**Effect of baicalin on lung tissue histopathological changes in LPS-induced ALI mice.** LPS + baicalin group mice (*n* = 6 per group) were given an intraperitoneal injection of baicalin (25, 50, and 100 mg/kg) 1 h prior to LPS administration. After 6 h, 12 h, and 24 h, lungs were harvested for measuring histopathological changes. To confirm pathological changes in lung tissues, hematoxylin–eosin (H&E) staining was performed: (**A**) Mice lung treated with LPS and baicalin for 6 h; (**B**) mice lung treated with LPS and baicalin for 24 h; (**C**) mice lung treated with LPS and baicalin for 12 h; the resolution was ×200.

**Figure 6 molecules-28-01873-f006:**
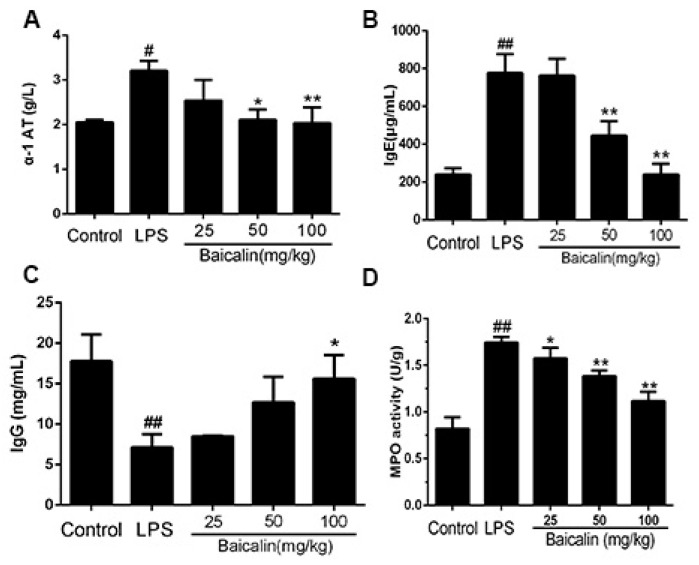
**Effect of baicalin on the contents of α-1 AT, IgE, IgG, and MPO in LPS-induced ALI mice.** LPS + baicalin group mice (*n* = 6 per group) were given an intraperitoneal injection of baicalin (25, 50, and 100 mg/kg) 1 h prior to LPS administration. After 12 h, the blood was collected. (**A**) α-1 AT content of each group; (**B**) IgE content of each group; (**C**) IgG content of each group; (**D**) MPO activities of each group. The values presented are the means ± SEM (*n* = 6 in each group). ^#^ *p* < 0.05 vs. control group; ^##^ *p* < 0.01 vs. control group; * *p* < 0.05 vs. LPS; ** *p* < 0.01 vs. LPS.

**Figure 7 molecules-28-01873-f007:**
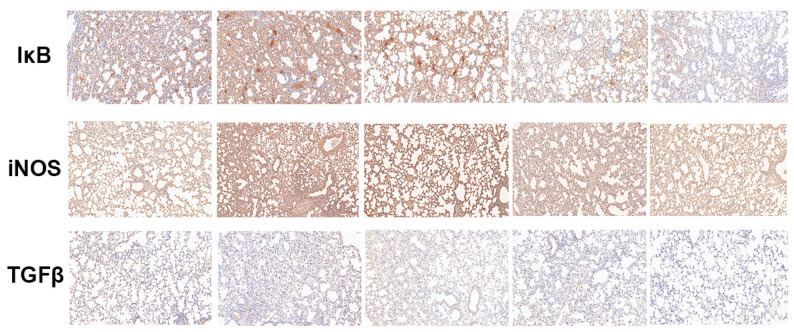
**Effect of baicalin on the protein expression level of IκB, TGF-β1, and iNOS in LPS-induced ALI mice.** LPS + baicalin group mice (*n* = 6 per group) were given an intraperitoneal injection of baicalin (25, 50, and 100 mg/kg) 1 h prior to LPS administration. After 12 h, the blood was collected. The resolution was ×200.

**Table 1 molecules-28-01873-t001:** Effect of baicalin on the number and proportion of leukocyte cell in LPS-induced mice.

	Treatment ^1^						
Items	CON	LPS (5 mg/kg)	LPS + BA (25 mg/kg)	LPS + BA (50 mg/kg)	LPS + BA (100 mg/kg)	SEM	*p*-Value
WBC ^2^ (10^9^/L)	6.1 ^b^	13.1 ^a^	8.6 ^b^	8.0 ^b^	6.0 ^b^	0.77	0.002
Lymph ^3^ (10^9^/L)	4.1 ^bc^	8.4 ^a^	5.1 ^b^	4.3 ^bc^	3.1 ^c^	0.52	0.000
Mon ^4^ (10^9^/L)	0.3 ^b^	1.3 ^a^	0.8 ^ab^	0.4 ^b^	0.3 ^b^	0.13	0.033
Gran ^5^ (10^9^/L)	1.7 ^b^	3.4 ^a^	2.7 ^a^	3.3 ^a^	2.6 ^ab^	0.20	0.023
Lymph%	67.73 ^a^	64.50 ^a^	59.73 ^ab^	53.77 ^b^	51.73 ^b^	1.93	0.009
Mon%	4.59	9.67	8.77	4.90	4.80	0.89	0.180
Gran%	27.63 ^bc^	25.87 ^c^	31.50 ^b^	41.33 ^a^	43.43 ^a^	1.99	0.000
RBC ^6^ (10^12^/L)	12.02	11.43	12.09	11.57	11.38	0.73	0.998
HGB ^7^ (g/L)	183	172	190	175	174	7.18	0.950
HCT ^8^%	58.0	54.0	58.3	57.1	55.9	1.79	0.961
MCV ^9^ (fL)	48.3	47.3	48.3	49.4	49.2	1.37	0.993
MCH ^10^ (pg)	15.2	15.0	14.8	15.1	15.2	0.73	1.000
MCHC ^11^ (g/L)	315	318	308	306	311	3.73	0.886
RDW ^12^%	14.3	13.5	14.3	13.3	14.0	0.56	0.980
PLT ^13^ (10^9^/L)	1143	1017	874	1072	1039	58.06	0.739
MPV ^14^ (fL)	7.2	6.7	6.9	6.8	6.6	0.41	0.995
PDW ^15^	16.7	16.9	16.8	16.7	16.6	0.66	1.000

^1^ CON: Control group; LPS: LPS group; LPS + BA: LPS + baicalin group. *n* = 6 in each group; ^2^ WBC: White blood cell; ^3^ Lymph: Lymphocyte; ^4^ Mon: Monocyte; ^5^ Gran: Neutrophil; ^6^ RBC: Red blood cell; ^7^ HGB: Hemoglobin; ^8^ HCT: Hematocrit; ^9^ MCV: Mean Corpuscular Volume; ^10^ MCH: Mean Corpuscular Hemoglobin;^11^ MCHC: Mean Corpuscular Hemoglobin Concentration; ^12^ RDW: Red cell distribution width; ^13^ PLT: Platelets; ^14^ MPV: Mean Platelet Volume; ^15^ PDW: Platelet distribution width. ^a−c^ Means with different superscript letter differ (*p* < 0.05) within groups.

## Data Availability

Not applicable.

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
