# Peer review of "Baicalin Relieves LPS-Induced Lung Inflammation via the NF-κB and MAPK Pathways"

_molecules, 2023, doi:10.3390/molecules28041873_

Round 1

Reviewer 1 Report

In this manuscript, the author investigated the effects of Baicalin on LPS-induced inflammation and the specific mechanism involved in mice and cells. They found that Baicalin inhibited ALI inflammation in mice and cells. However, still more experiments are required to address this question.

As I mentioned previously, the author should discuss the originality of the manuscript

The following published studies are very similar to the present study.

C. Jiang et al. Baicalein suppresses lipopolysaccharide-induced acute lung injury by regulating Drp1-dependent mitochondrial fission of macrophages. Biomedicine & Pharmacotherapy 145 (2022) 112408. https://doi.org/10.1016/j.biopha.2021.112408

Duan et al. Baicalin attenuates LPS-induced alveolar type II epithelial cell A549 injury by attenuation of the FSTL1 signaling pathway via increasing miR-200b-3p expression. Innate Immunity 27(4) https://doi.org/10.1177/17534259211013887

Bao et al. Baicalin Alleviates LPS-Induced Oxidative Stress via NF-κB and Nrf2–HO1 Signaling Pathways in IPEC-J2 Cells. Front. Vet. Sci., 25 January 2022. https://doi.org/10.3389/fvets.2021.808233

Gao et a.l Mechanistic insights into the amelioration effects of lipopolysaccharide-induced acute lung injury by baicalein: An integrated systems pharmacology study and experimental validation. Pulmonary Pharmacology & Therapeutics Volumes 73–74, June 2022, 102121 https://doi.org/10.1016/j.pupt.2022.102121

The author should measure interesting traments like exposure or time points  in specific mice cell populations. These findings will strongly support the potential use of lung-delivered therapy of Baicalin in human applications, particularly in a short-term approach following untoward and extreme occupational and/ or environmental inhalation exposures.

As I mentioned above the author should measure interesting exposure or time points, specific cell population in mice. For example Baicalin treatment following a one‑time high-dose LPS exposure modulates specific lung monocyte/macrophage populations. Effect of Baicalin treatment following a one‑time high dose LPS exposure in modulating neutrophils and lymphocyte lung populations.

The current format in methods could be more straightforward; authors should rewrite the generation of the LPS model with the exact dose of LPS per body weight of mice. Spices types like C57BL/6 or BALB c etc.

The conclusions  are not consistent with the evidence and arguments presented. It is important, if they add in-vivo experiments with specific populations, such as recruitment/ transitioning monocyte/macrophage populations or neutrophil recruitment. I hope it will address present study question and useful develop lung-delivered therapy of Baicalin in human applications

Authors should add more appropriate references, introduction and discussion. 

Fig.5 A-E, the author needs to improve the figure quality; the scale bar is missing. The author must write a detailed explanation of the changes in the lung histology of LPS-induced mice with Baicalin. Authors must improve with other immunostaining for immune cells, adhesion and inflammatory markers in the lung of LPS-induced mice with Baicalin.

Fig.5 authors put the star in fig5F and other bar graphs; authors must include a line/bar on the top of comparing two groups.

Fig.2 author must include a scale bar. All western blot need to be quantified. Author must include a dot plot instead bar graph. The author must include the number of experiments/mice and replication in the figure legends.

Author Response

  1. As I mentioned previously, the author should discuss the originality of the manuscript

The following published studies are very similar to the present study.

  1. Jiang et al. Baicalein suppresses lipopolysaccharide-induced acute lung injury by regulating Drp1-dependent mitochondrial fission of macrophages. Biomedicine & Pharmacotherapy 145 (2022) 112408. https://doi.org/10.1016/j.biopha.2021.112408

Duan et al. Baicalin attenuates LPS-induced alveolar type II epithelial cell A549 injury by attenuation of the FSTL1 signaling pathway via increasing miR-200b-3p expression. Innate Immunity 27(4) https://doi.org/10.1177/17534259211013887

Bao et al. Baicalin Alleviates LPS-Induced Oxidative Stress via NF-κB and Nrf2–HO1 Signaling Pathways in IPEC-J2 Cells. Front. Vet. Sci., 25 January 2022. https://doi.org/10.3389/fvets.2021.808233

Gao et a.l Mechanistic insights into the amelioration effects of lipopolysaccharide-induced acute lung injury by baicalein: An integrated systems pharmacology study and experimental validation. Pulmonary Pharmacology & Therapeutics Volumes 73–74, June 2022, 102121 https://doi.org/10.1016/j.pupt.2022.102121

Response: Thank you for your question. Our manuscript mainly revealed the mechanism of baicalin on alleviating inflammation in LPS- induced acute lung injury mice and RAW 264.7 macrophage cell via NF-κB and MAPK pathways. Our main focus is on the angle of reducing inflammation and regulating the release of inflammatory cytokines.

Jiang et al.(2022) mainly expounded the mechanism of baicalin suppressing acute lung injury from the point of view of regulating mitochondrial disorder. And it is using bone marrow-derived macrophages (BMDMs).

Duan et al. (2021) stated that baicalin could significantly inhibit the expression of inflammation-related proteins and improve LPS-induced inflammatory injury in alveolar type II epithelial cells, whose mechanism may be related to increasing the expression of miR-200b-3p. In addition, the main content of this study is to reveal that FSTL1 is a regulation target of miR-200b-3p.

Bao et al.(2022) mainly explained the mechanism of baicalin inhibiting LPS-induced oxidative stress and protecting the normal physiological function of IPEC-J2 cells via NF-κB and Nrf2-HO1 signaling pathways. The main idea is to reveal how baicalin attenuated LPS-induced oxidative stress in Porcine IECs (IPEC-J2 cells).

Gao et a.l (2022)’s purpose is mainly through an integrated systems pharmacology study to validate the effect of baicalin. The main experimental means used were the evaluation of ADME, collection of acute lung injury-related targets, selection of latent targets for baicalein against acute lung injury, establishment of protein–protein interaction (PPI) network and collection of key targets, gene ontology (GO) biolog-ical process and the kyoto encyclopedia of genes and genomes (KEGG) enrichment analyses, screening out latent targets of baicalein against pyroptosis. The ultimate goal of this experiment is to verify the target of baicalin obtained by other means.

  1. The author should measure interesting traments like exposure or time points in specific mice cell populations. These findings will strongly support the potential use of lung-delivered therapy of Baicalin in human applications, particularly in a short-term approach following untoward and extreme occupational and/ or environmental inhalation exposures.

Response: Thank you very much for your suggestion. We had supplemented the different time points of baicalin treatment in mice. The mice in LPS and LPS with baicalin were treated for 6h, 12h and 24h. The serum were collected for detecting the number of WBC, lymph, monocyte and neutrophil. The lung was collected for pathological analysis. The results showed that baicalin treatment for 6h, 12h and 24h could be mitigated the number of WBC and 12h treatment achieved the best results (Table.1). Meanwhile, LPS induced for 6h, 12h and 24h could both cause inflammatory pathological changes in the lung, such as alveolar narrowing, lung space widening, inflammatory cell enrichment, congestion, etc. Baicalin treatment for 6h, 12h and 24h both can significantly alleviate these pathological changes, and 12h remission was best. This suggested that baicalin should be treated for 12 hours to achieve the best effect (Fig.6).

  1. As I mentioned above the author should measure interesting exposure or time points, specific cell population in mice. For example Baicalin treatment following a one‑time high-dose LPS exposure modulates specific lung monocyte/macrophage populations. Effect of Baicalin treatment following a one‑time high dose LPS exposure in modulating neutrophils and lymphocyte lung populations.

Response: Thank you very much for your suggestion. White blood cells (WBC) are the "guardian" of the human body in the fight against disease. When the bacteria invade the body, WBC can pass through the capillary wall through deformation, concentrate on the invasion site of the bacteria, and surround and phagocytose the bacteria. If the number of WBC in the body is higher than normal, it is likely that the body has inflammation.We had supplemented the data of different time points of baicalin and LPS treatment in mice. We determined changes in the number of various cells in the blood of mice at different treatment times. The results showed that the number of WBC (including lymphocyte, monocyte and neutrophil) increased most at 12 h after LPS treatment compared with the control group. Meanwhile, baicalin treatment for 12h resulted in the best reduction of inflammatory cells compared with 6h and 24h. These results suggest to us that the optimal time to use baicalin quality after exposure to high doses of LPS is 12 h.

  1. The current format in methods could be more straightforward; authors should rewrite the generation of the LPS model with the exact dose of LPS per body weight of mice. Spices types like C57BL/6 or BALB c etc.

Response: Thank you very much for your suggestion. We had revised the LPS dose into “5mg/kg”, per body weight of mice is 0.1 mg. The species type of mice had been stated in Cell culture and animal feeding Liaoning Changsheng Biotechnology offered male BALB/c mice weighing between 18 and 22 g on Line 75.

  1. The conclusions are not consistent with the evidence and arguments presented. It is important, if they add in-vivo experiments with specific populations, such as recruitment/ transitioning monocyte/macrophage populations or neutrophil recruitment. I hope it will address present study question and useful develop lung-delivered therapy of Baicalin in human applications.

Response: Thank you very much for your suggestion. We had added the different time of baicalin treatment. The results showed that Baicalin used 12 hours after LPS exposure achieved the best effect of anti-inflammatory cell enrichment. Detailed data are provided in Supplementary Table 1.

6.Authors should add more appropriate references, introduction and discussion.

Response: Thank you very much for your suggestion. We had added more references in introduction and discussion.

  1. 5 A-E, the author needs to improve the figure quality; the scale bar is missing. The author must write a detailed explanation of the changes in the lung histology of LPS-induced mice with Baicalin. Authors must improve with other immunostaining for immune cells, adhesion and inflammatory markers in the lung of LPS-induced mice with Baicalin.

Response: Thank you very much for your suggestion. Sorry for making such mistakes. We had added the scale bar in Fig.5. And according to your suggestion, we had remade the H&E stained sections. Meanwhile, we had added the different exposure time of LPS and baicalin. For 12h, the lung of LPS group showed that the alveolar septum was thickened, the alveolar wall telangiectasia was congested, and a large number of inflammatory cells infiltrated near the bronchi (deep blue purple). Neutrophils and macrophages could be seen in the alveolar space. And after baicalin treatment for 6h, 12h and 24h, the infiltration of inflammatory cells was significantly reduced, and the alveolar structure was intact. The degree of inflammatory cell infiltration, capillary congestion, hemorrhage and alveolar wall thickening were gradually reduced in the baicalin treatment for 12h groups.

  1. 5 authors put the star in fig5F and other bar graphs; authors must include a line/bar on the top of comparing two groups.

Response: Thank you for your suggestion. In the figure legends, we had expound the meaning of #,##,* and **. # indicates P<0.05, which means significant differences from the control group, ##P<0.01 vs. Control; *P<0.05 vs. LPS, **P<0.01 vs. LPS.

  1. 2 author must include a scale bar. All western blot need to be quantified. Author must include a dot plot instead bar graph. The author must include the number of experiments/mice and replication in the figure legends.

Response: Thank you for your suggestion. Sorry for making such mistake, and we had added the scale bar in Fig.2. All the western blot had already be quantified and were represented by a bar graph. We had added the number of experiments mice in the figure legends of Fig.4 and 5. “LPS+baicalin group mice (n=6 per group) were given an intraperitoneal injection of baicalin 1 h prior to administration of LPS.”.

Reviewer 2 Report

The paper titled: Baicalin relieves the LPS-induced lung inflammation via NF-κB and MAPK pathways has been reviewed and its contents are very good and will help in ameliorating acute lung injury through the use of safe natural substances through the control of different signaling pathways such as TLR4, NFkB and inflammatory cytokines, together with some transcriptional factors.

I need authors to reply these comments to get final acceptance for their valuable data.

1- Abstract: need more detailed description as it it is very short in its contents

2- In materials and methods:

a) In cell viability assay, based on what you used the different doses of bacialin ( cite references)

b) In western blot analysis: why you used bacialin doses (60, 90, and 20 ug/ml)?, How about other doses (150 and 180)?

c) In page 2 line 91, What are these substances? give clarification

d) In page 3 line 99: 80 ug protein per lane, Is it high? and what the percentage of SDS-PAGE used?

e) In vivo study: give reference for LPS and bacialin doses used.

f) Why you didn't measure some lung inflammatory markers  such as CP, alpha-1 antitrypsin, TgE and IgG. it is better to add to strength your hypothesis.

g) Why you didn't collect heparinized blood to examine the change in CBC, differential leukocyte counts and other mediators

h) histopathology need some extra data such as as immunohistochemistery for some genessuch as NFkB, TGF-b1 and iNOS.

Results:

1) What the effect of bacialin alone on different examined cytokines

2) Page 5 line 180 Correct the sentence to be: pre-treatment with bacialin after LPS inhibited LPS-induced upregulation in phosphorylation of IkB alpha,.......etce

Discussion well written

You need to add conclusion section

You need to add abbreviation list

Some minor points can bee seen in attached file

Author Response

The paper titled: Baicalin relieves the LPS-induced lung inflammation via NF-κB and MAPK pathways has been reviewed and its contents are very good and will help in ameliorating acute lung injury through the use of safe natural substances through the control of different signaling pathways such as TLR4, NFkB and inflammatory cytokines, together with some transcriptional factors.

I need authors to reply these comments to get final acceptance for their valuable data.

  • Abstract: need more detailed description as it it is very short in its contents

Response: Thank you for your suggestion. We had added details in abstract.

2- In materials and methods:

  1. In cell viability assay, based on what you used the different doses of bacialin ( cite references)

Response: Thank you for your suggestion. We used 60, 90, 120, 150 and 180 μg/ml to detect the effect of baicalin on cell viability. We found that all the concentrations were noncytotoxic to RAW 264.7 cells. Li et.al (2020) examined the toxicity of baicalin (10, 25, 50, 100, 150 μg/mL) on Th-17 cells and Treg cells. The results showed that the viability of Th17 and Treg cells was decreased by 150 μg/mL baicalin treatment. Although our results showed that 150 μg/ml was not toxic, we chose 60, 90, 120 μg/ml in order to reduce the effect on cell viability in the following experiment.

Reference

[1]Li J, Lin X, Liu X, Ma Z, Li Y. Baicalin regulates Treg/Th17 cell imbalance by inhibiting autophagy in allergic rhinitis. Mol Immunol. 2020 Sep;125:162-171. doi: 10.1016/j.molimm.2020.07.008. Epub 2020 Jul 17. PMID: 32688118.

  1. b) In western blot analysis: why you used bacialin doses (60, 90, and 20 ug/ml)?, How about other doses (150 and 180)?

Response: Thank you for your suggestion. In a), we had expounded that the reason we chose 60,90, and 120 μg/ml. And the supplement results showed that there was no significant change in each protein after application of 150 and 180 μg/ml. So we only presented the results for 60, 90, and 120 μg/ml in the main text section.

  1. c) In page 2 line 91, What are these substances? give clarification

Response: Thank you for your suggestion. “Pyrrolidine dithiocarbamate (PDTC, an NF-κB inhibitor), SP600125 (a JNK inhibitor), SB203580 (a p38 MAPK inhibitor), U0126, (an ERK inhibitor)” had been expounded in Chemicals and reagents on Line 89 and 90.

  1. d) In page 3 line 99: 80 ug protein per lane, Is it high? and what the percentage of SDS-PAGE used?

Response: Thank you for your suggestion.To get a clearer WB image,, so we applied a maximum of 80 μg of protein. It is appropriate for our results. The percentage of SDS-PAGE used was 10%, and we would added the detail in the manuscript.

  1. e) In vivo study: give reference for LPS and bacialin doses used.

Response: Thank you for your suggestion. Actually we had carried out pre-experiment to detect the accurate dose of LPS. We use LPS 1 mg/kg, 5 mg/kg  and 10 mg/kg and results showed that 1 mg/kg of LPS could only minor damage to the lungs, and 10 mg/kg of LPS caused death in mice within 24 h. So we chose 5 mg/kg as induced dose which was in agreement with the study by Zhu et al [1].

Zhao et, al. used 50 mg/kg baicalin to detect the effect on bleomycin-induced pulmonary fibrosis and fibroblast proliferation in rats[2], and Zheng et, al. used 100 mg/kg to detect the impact of baicalin on autophagy-dependent ferroptosis in early brain injury after subarachnoid hemorrhage [3]. Referring to their study, we chose the three concentrations of 25, 50 and 100 mg/kg as the treatment doses.

Reference

  • Zhu P, Wang J, Du W, Ren J, Zhang Y, Xie F, Xu G. NR4A1 Promotes LPS-Induced Acute Lung Injury through Inhibition of Opa1-Mediated Mitochondrial Fusion and Activation of PGAM5-Related Necroptosis. Oxid Med Cell Longev. 2022 Feb 18;2022:6638244.
  • Zhao H, Li C, Li L, Liu J, Gao Y, Mu K, Chen D, Lu A, Ren Y, Li Z. Baicalin alleviates bleomycin‑induced pulmonary fibrosis and fibroblast proliferation in rats via the PI3K/AKT signaling pathway. Mol Med Rep. 2020 Jun;21(6):2321-2334.
  • Zheng B, Zhou X, Pang L, Che Y, Qi X. Baicalin suppresses autophagy-dependent ferroptosis in early brain injury after subarachnoid hemorrhage. Bioengineered. 2021 Dec;12(1):7794-7804.
  1. f)Why you didn't measure some lung inflammatory markers such as CP, alpha-1 antitrypsin, TgE and IgG. it is better to add to strength your hypothesis.

Response: Thank you for your suggestion. We had added some data, such as alpha-1 antitrypsin (α-1 AT), IgE and IgG. 

α1-AT is one of the most important protease inhibitors synthesized by hepatocytes in vivo. Due to its low molecular weight, it is easy to diffuse in lung tissue and combine with destructive proteolytic enzymes, which can prevent lung tissue from being damaged by proteolytic enzymes, thereby protecting the normal structure and function of lung tissue. In addition, α1-AT is also an acute phase protein. When the body has inflammation, infection, cachexia and other conditions, its blood concentration will increase 3 to 4 times. Our results showed that after LPS induced, the content of α-1 AT increased significantly compared with control group, which illustrated that during the acute lung injury, elastase and trypsin are released in large quantities of mice, resulting in an imbalance of the ratio of protease to α-1 AT. The results are consistent with Knoell’s (1998) [1]and Jie’s studies [2] . And applicant of baicalin (25, 50, 100 mg/kg) reduced the increasing content of α-1 AT and showed a dose-dependent manner.

IgE is mainly synthesized by B cells in the lymphoid tissue of the lamina propria of the mucosa of the respiratory tract and digestive tract, which is a mediator of allergic reactions. It is a secreted immunoglobulin that is the main antibody causing type I hypersensitivity and has immune functions capable of binding to mast cells and basophils. Meanwhile, LPS-treatment induced mice produces higher IgE level compared with control group, while baicalin significantly reduced the production of this allergena in a dose-dependent manner. The results are consistent with Dong’s (2021) study [3].

IgG is the main component of serum immunoglobulin, which plays a protective role in the body's immunity and responds to most antibacterial and antiviral drugs. It can effectively prevent the corresponding infectious diseases by coping with most bacteria, viruses, measles, hepatitis A, etc. The content of serum IgG was detected simultaneously. Mice in control group had a higher level of IgG compared with LPS group. And three concentration of baicalin (25, 50, 100 mg/kg) restored the IgG level gradually.

Reference

  • Knoell DL, Ralston DR, Coulter KR, Wewers MD. Alpha 1-antitrypsin and protease complexation is induced by lipopolysaccharide, interleukin-1beta, and tumor necrosis factor-alpha in monocytes. Am J Respir Crit Care Med. 1998 Jan;157(1):246-55.
  • [2] Zhijun J, Wenlan Y, Yingyun C et,al. Protective EFFECT OF α-1 anti trypsin on LPS-induced acute lung injury in rabbits [J]. Chin J Tuberculosis & Respiratory. 2000(04):250-251.
  • Dong J, Xu O, Wang J, Shan C, Ren X. Luteolin ameliorates inflammation and Th1/Th2 imbalance via regulating the TLR4/NF-κB pathway in allergic rhinitis rats. Immunopharmacol Immunotoxicol. 2021 Jun;43(3):319-327.
  1. g) Why you didn't collect heparinized blood to examine the change in CBC, differential leukocyte counts and other mediators

Response: Thank you for your suggestion. We determined changes in the number of various cells in the blood of mice following baicalin treatment for 6, 12, and 24h. The results showed that compared with the control group, the number of WBCs (including lymphocytes, monocytes, and neutrophils) increased most at 12 h after LPS induction. Meanwhile, baicalin treatment for 12h resulted in the strongest reduction in inflammatory cells compared with at 6 and 24h (Table 1 and Supplementary Table 1). h) histopathology need some extra data such as as immunohistochemistery for some genessuch as NFkB, TGF-b1 and iNOS.

Response: Thank you for your suggestion. We had added the extra data. Immunohistochemical technique was applied to detect the expression of IκB, iNOS and TGF-β. The results showed that after LPS induced, the protein expression of IκB, iNOS and TGF-β were increased compared with the control mice. The baicalin pre-treatment reduced the ascending protein expression (Fig.7).

Results:

1) What the effect of bacialin alone on different examined cytokines

Response: Thank you for your suggestion. Actually, we detected the baicalin alone on different examined cytokines in the pre-experiment, and the results showed that there was no significantly change on the contents of TNF-α,IL-1β and IL-6 compared with the control group. It is suggested that baicalin alone had no effect on inflammatory cytokines.

2) Page 5 line 180 Correct the sentence to be: pre-treatment with bacialin after LPS inhibited LPS-induced upregulation in phosphorylation of IkB alpha,.......etce

Response: Thank you for your suggestion. We felt very sorry for making such mistakes and had corrected it in the manuscript.

Discussion well written

You need to add conclusion section

Response: Thank you for your suggestion. We had added the conclusion section.

You need to add abbreviation list

Response: Thank you for your suggestion. We had added the abbreviation list.

Some minor points can bee seen in attached file

Response: Thank you for your suggestion.

  • The meaning of W/D ratio.

Response: Thank you for your suggestion. The Wet/Dry ratio could intuitively indicates the degree of edema in lungs. The measure methods was on Line165-168.

  • We had rewrited this sentence on Line 213.

Round 2

Reviewer 1 Report

The authors have satisfactorily responded to all comments and made the necessary changes to the manuscript.

Reviewer 2 Report

I have checked the new added data and I think all my comments have been answered by authors.
I grauntee the acceptance of the paper in its current form .